Vitamin C supplementation reduces expression of circulating miR-451a in subjects with poorly controlled type 2 diabetes mellitus and high oxidative stress

Ruknarong Laongthip 1 2 3
Boonthongkaew Chongchira 2 3
Chuangchot Nisa 1 2 3
Jumnainsong Amonrat 1 4
Leelayuwat Naruemon 3 5
Jusakul Apinya 1 4
Gaudieri Silvana 6 7 8
Leelayuwat Chanvit chanvit@kku.ac.th 1 4
1 Centre for Research and Development of Medical Diagnostic Laboratories (CMDL), Faculty of Associated Medical Sciences, Khon Kaen University , Khon kean , Thailand
2 Biomedical Sciences Program, Graduate School, Khon Kaen University , Khon Kaen , Thailand
3 Exercise and Sport Sciences Development and Research Group (ESRG), Faculty of Medicine, Khon Kaen University , Khon Kaen , Thailand
4 Department of Clinical Immunology and Transfusion Sciences, Faculty of Associated Medical Sciences, Khon Kaen University , Khon Kaen , Thailand
5 Graduate School, Khon Kaen University , Khon Kaen , Thailand
6 School of Human Sciences, University of Western Australia , Perth , Western Australia , Australia
7 Institute for Immunology and Infectious Diseases, Murdoch University , Perth , Western Australia , Australia
8 Division of Infectious Diseases, Department of Medicine, Vanderbilt University Medical Center, Vanderbilt University , Nashville , TN , United States of America
Davalos Alberto
Electronic publication date: 2021 Feb 4
Publication date: 2021
Volume: 9
Electronic Location ID: e10776
Received 2020 Jul 21; Accepted 2020 Dec 22
Copyright: ©2021 Ruknarong et al.
Copyright year: 2021
Copyright holder: Ruknarong et al.
License: This is an open access article distributed under the terms of the Creative Commons Attribution License, which permits unrestricted use, distribution, reproduction and adaptation in any medium and for any purpose provided that it is properly attributed. For attribution, the original author(s), title, publication source (PeerJ) and either DOI or URL of the article must be cited.
License URL: https://creativecommons.org/licenses/by/4.0/

Keywords: Vitamin C, miRNA, Type 2 diabetes, Oxidative stress, miRNA array

Funding: The Royal Golden Jubilee (RGJ) Ph.D. Programme PHD/0045/2557 This work was supported by The Royal Golden Jubilee (RGJ) Ph.D. Programme (PHD/0045/2557). The funders had no role in study design, data collection and analysis, decision to publish, or preparation of the manuscript.

==============================
Background

Vitamin C is an essential element required for normal metabolic function. We investigated the effect of vitamin C supplementation on circulating miRNA (miR) expression in subjects with poorly controlled type 2 diabetes mellitus (T2DM). Changes in miR expression were also correlated with clinical measures of disease.

Methods

Pre- and post-vitamin C supplementation samples from five participants who had increased vitamin C levels, improved oxidative status and polymorphonuclear (PMN) function after receiving 1,000 mg of vitamin C daily for six weeks were screened for miRNA expression using the NanoString miRNA assay. Differences in miRNA expression identified from the miRNA screen were validated by qRT-PCR.

Results

Four miRNAs showed significantly different expression post-vitamin C supplementation relative to baseline, including the down-regulation of miR-451a (−1.72 fold change (FC), p = 0.036) and up-regulation of miR-1253 (0.62 FC, p = 0.027), miR-1290 (0.53 FC, p = 0.036) and miR-644a (0.5 FC, p = 0.042). The validation study showed only miR-451a expression was significantly different from baseline with vitamin C supplementation. MiR-451a expression was negatively correlated with vitamin C levels (r =  − 0.497, p = 0.049) but positively correlated with levels of malondialdehyde (MDA) (r = 0.584, p = 0.017), cholesterol (r = 0.564, p = 0.022) and low-density lipoproteins (LDL) (r = 0.522, p = 0.037). Bioinformatics analysis of the putative target genes of miR-451a indicated gene functions related to signaling pathways involved in cellular processes, such as the mammalian target of rapamycin (mTOR) signaling pathway.

Conclusions

Vitamin C supplementation altered circulating miR-451a expression. The results from this pilot study suggest that miRNAs could be used as biomarkers to indicate oxidative status in subjects with T2DM and with poor glycemic control and could lead to a novel molecular strategy to reduce oxidative stress in T2DM.

Introduction

Vitamin C, ascorbic acid or ascorbate, is well known as an antioxidant required for normal metabolic function of the body and is associated with a wide spectrum of biological processes (Mandl, Szarka & Banhegyi, 2009). As an indication of the protective properties of vitamin C, studies have shown that vitamin C can diminish the detrimental effects of hyperglycemia such as oxidative stress, inflammation as well as endothelial dysfunction (Ceriello et al., 2013). Several studies have also reported a beneficial effect of vitamin C supplementation in type 2 diabetes mellitus (T2DM); a chronic metabolic disease associated with chronic hyperglycemia and excess free fatty acids (FFAs) (Hawkins et al., 2003). This complex disorder is characterized by insulin resistance in which the pancreas produces insufficient insulin or there is an ineffective response (American Diabetes, 2018). The pathophysiology of T2DM is not fully understood but high oxidative stress and low-grade inflammation are thought to be involved and can exacerbate the development and progression of the disease. Overproduction of reactive oxygen species (ROS) during the course of T2DM can cause DNA and protein damage, lipid peroxidation, cellular and vascular dysfunction leading to diabetic complications that have been associated with high levels of oxidative stress, as indicated by the markers malondialdehyde (MDA) and F2-Isoprostanes (F2IsoPs), and low levels of vitamin C (Bhatia et al., 2003; Johansen et al., 2005; Kaviarasan et al., 2009). Moreover, impaired immune responses, such as neutrophil function, have been reported in subjects with T2DM (Chanchamroen et al., 2009).

There is evidence that daily supplementation with 1000 mg/day of vitamin C for six weeks can improve blood glucose, lipid profiles, glycated haemoglobin (HbA1C) and insulin in subjects with T2DM (Afkhami-Ardekani & Shojaoddiny-Ardekani, 2007). The effect of vitamin C on subjects can be observed at the gene expression level and mechanistically may involve epigenetic regulation including via microRNAs (miRNAs or miRs) (Kolhe et al., 2018). Supporting this model, Kim et al. (2015) revealed dietary consumption of high dose vitamin C enhances anti-oxidation and anti-glycation as well as reduction of inflammation-related miRNAs in lipoproteins. Furthermore, treatment of human bone marrow stem cells with vitamin C has been shown to modulate miRNA expression in these cells (Kolhe et al., 2018). These studies strongly suggest that vitamin C can affect miRNA expression and may be involved in the pathophysiology of T2DM.

MiRNAs are functional non-coding small RNAs typically 18–22 nucleotides (nt) in length that have been identified as important regulators of gene expression. These RNA molecules mainly act by repressing mRNA translation at the 3′ untranslated region (3′UTR) of the target mRNA and can function in cell–cell communication (Bartel, 2004; Turchinovich, Tonevitsky & Burwinkel, 2016). One of the important properties of these small RNAs is that they are highly stable in various sample types such as serum or plasma and, accordingly, can be used as a biomarker or as a target for therapeutics for several diseases including T2DM, in which miRNA deregulation has been associated with progression of the disease (He et al., 2017; Jimenez-Lucena et al., 2018; Maqbool & Ul Hussain, 2014; Regazzi, 2018). However, less is known about the effect of vitamin C supplementation on miRNAs in subjects with T2DM, especially for those subjects with poor glycemic control. In such cases, the high oxidative stress and low levels of plasma vitamin C found in these subjects may be influenced by miRNAs and vice versa.

To determine the effect of vitamin C supplementation on miRNA expression in subjects with T2DM, we investigated circulating miRNAs before and after vitamin C supplementation using a miRNA screen. We restricted the analysis to those subjects that showed changes in plasma vitamin C levels, oxidative stress and polymorphonuclear (PMN) function following vitamin C supplementation. We hypothesized that the expression level of miRNAs may be affected by vitamin C supplementation and these changes may correlate with clinical measures known to reflect the pathophysiological processes in T2DM. Such associations may provide greater insights into the molecular mechanisms involved in these biological processes and may be used as biomarkers for responses to the dietary treatment of the disease leading to novel strategies in molecular targeting of relevant genes in T2DM.

Materials & Methods

Study design and participants

Archived plasma samples from a subset of participants who had received vitamin C as part of a study examining vitamin C supplementation in subjects with T2DM (Chuangchot et al., 2020) were utilized in this study and their usage was approved by the Human Research Ethics Committee of Khon Kaen University in accordance with the 1964 Declaration of Helsinki (HE571264). De-identified diagnostic measures of T2DM and clinical data of the subjects were received from Srinagarind Hospital, Faculty of Medicine, Khon Kaen, Thailand. The selection criteria for subjects in the larger study included age between 30 and 60 years old, diagnosed as having T2DM according to type 2 diabetes diagnostic criteria by the American Diabetes Association (ADA) for at least 12 months, poor glycemic control (defined by HbA1c ≥8.5% or 69.4 mmol/mol) without diabetic complications, treated with oral diabetic drugs only (Metformin and/or Glibenclamide), sedentary and lack of any regular exercise program for at least 6 months, and living in the Khon Kaen Province, Thailand. Some subjects were treated with lipid-lowering drugs and/or anti-hypertensive drugs for the duration of the study to maintain blood pressure at ≤ 140/90 mmHg. Subjects with any other diagnosed chronic disease including neuromuscular disorder, liver and kidney disease, positive chronic infection including human immunodeficiency virus, hepatitis and tuberculosis, abnormal resting electrocardiogram, changes in medical treatment during the study period, insulin injection, blood pressure higher than 140/90 mm Hg, and smokers were excluded from the study.

The participants received 1,000 mg/day vitamin C for six weeks. Overnight fasting blood samples were collected at pre-and post-vitamin C supplementation. Biochemical measurements were determined as follows: HbA1c by Cobas c501 (Roche Diagnostics, Mannheim, Germany); lipid profile by Reflotron Plus (Boehringer Mannheim, Mannheim, Germany); fasting blood sugar by a glucose analyzer (YSI 2300 STAT PLUS, Champaign, IL 61822, USA); and insulin by an immunoradiometric assay kit ((MP Biomedical Germany GmbH, Eschwege, Germany). Plasma vitamin C levels were measured as previously described (Zhang et al., 2009). PMN phagocytosis and oxidative burst were determined using flow cytometry as previously reported (Chuangchot et al., 2020). Plasma MDA concentration was examined using the thiobarbituric acid reactive substances (TBARS) assay (Draper et al., 1993). F2IsoPs concentration was determined using the direct 8-iso-Prostaglandin F2α Enzyme Immunoassay kit (EIA, 8-iso-PGF2α kit, Cayman Chemical Co). Blood samples from subjects with the following characteristics were used in this study: increased plasma vitamin C levels; increased phagocytosis and oxidative burst; and reduced oxidative stress as indicated by MDA and F2IsoPs concentrations at the post-vitamin C supplementation timepoint compared to pre-supplementation. Five participants were selected for the screening cohort and then an additional three participants were used for the validation cohort study (n = 8).

miRNA extraction

Total RNA was extracted from archived plasma using the miRNeasy Serum/Plasma kit (Qiagen, Germany) according to the manufacturer’s instruction. In brief, 200 uL of the plasma was mixed with five volumes of Qiazol and a fold of chloroform. Spike-in controls, cel-miR-39 and cel-miR-254, were added for normalization and as an indicator of extraction efficiency. After absorption, 14 uL of RNase-free water was added to the spin column to elute total RNA. MiRNA concentration was measured by a Nanophotometer (IMPLEN NanoPhotometer N60, Munich, Germany).

miRNA analysis

To examine changes in miRNA expression in response to vitamin C supplementation, a miRNA microarray screen was performed on the plasma samples from the five subjects using the NanoString platform (NanoString Technologies, Seattle, WA, USA). The miRNA screen includes 800 human mature-miRNAs and expression levels can be evaluated without an amplification step, thereby reducing the risk of PCR bias. Briefly, 3 uL of total RNA was ligated to specific miR-tags and hybridized to a color-coded probe. Counting of miRNA species and digital analysis of the resulting counts were performed using the nCounter human v3 miRNA expression platform. A conservative background threshold of 100 counts was set to remove targets with low expression levels from further analysis. Six positive controls and cel-miR-254 were used for data normalization.

qRT-PCR

Quantitative real-time polymerase chain reaction (qRT-PCR) was performed to validate miRNA expression levels for a select set of miRNAs in eight subjects with T2DM. The small RNAs were converted to cDNA using the miScript II RT kit (Qiagen, Germany) following the manufacturer’s protocol. The miScript primer assay and the miScript SYBR green PCR kit from Qiagen were used to measure the specific miRNAs on a Bio-Rad CF96™ real time PCR machine (Bio-Rad, Hercules, CA, USA) according to the manufacturer’s protocols. The cel-miR-39 spike-in control was used as a reference gene. Ct values >35 were deemed to be below background levels. Relative expression of individual miRNAs was calculated by the 2−ΔCt method.

Target gene prediction and pathway analysis

To identify the target genes of miR-451a, two bioinformatics tools Tarbase v. 8.0 and miRTarBase v. 8.0 were used and both tools have been developed based on experimental studies. The biological functions and enrichment pathways of these target genes were analyzed using the Database for Annotation, Visualization and Integration Discovery (DAVID) v. 6.8. A false discovery rate (FDR) with an adjusted P-value < 0.05 was set as the significance threshold.

Statistical analysis

Normal distribution was tested using the Shapiro–Wilk test. Log2 transformed data by nSolver Analysis Software v. 4.0 (NanoString Technologies) was used to analyze the expression of each miRNA with the visual genomics analysis studio (VGAS) program (http://www.iiid.com.au/software/vgas). Comparison of the pre- and post-vitamin C supplementation data was performed using a paired t-test. Laboratory measures and values from the validation study were compared between pre-and post-supplementation by paired t-test and Wilcoxon signed ranks test using SPSS statistics v. 19 (SPSS Inc., Chicago, IL, USA). A power of test (1-β) was checked by post-hoc analysis using G*Power v. 3.1.9.2 (Heinrich-Heine-Universität Düsseldorf, Germany). Spearman correlation was used to analyze the correlation between circulating miRNA expression and blood parameters and the figures were generated using GraphPad Prism v. 5.0 (GraphPad Software Inc., CA, USA). A p-value of <0.05 was set as the significance threshold. All data expressed as mean ± SD.

Results

Characteristics of subjects at pre- and post-vitamin C supplementation timepoints

Subjects with T2DM in the current study had an average age of 59 years, BMI of 26 kg/m2, HbA1c of 95 mmol/mol or 10.8% and duration of disease of 9 years (Table 1). Compared to baseline pre-vitamin C supplementation levels, subjects showed significantly increased plasma vitamin C levels, raised PMN phagocytosis and oxidative burst, reduced products of lipid peroxidase (MDA and F2IsoPs), as well as reduced levels of cholesterol following daily vitamin C supplementation for six weeks (Table 1).

Table 1 Characteristics of the participants at the pre-and post-vitamin C supplementation timepoints.

Parameters		Pre-supplementation
(n = 8)		Post-supplementation
(n = 8)		P-value	
Gender (female/male)		7/1		7/1		–	
Age (year)		58.8 ± 5.9		58.8 ± 5.9		–	
Duration of the disease (year)		9.3 ± 6.6		9.3 ± 6.6		–	
BMI (Kg/m2)		26.2 ± 4.1		26.2 ± 3.9		0.861	
HbA1c (mmol/mol)		95.0 ± 15.0		95.0 ± 14.8		0.909	
FBS (mmol/L)		13.0 ± 4.3		10.0 ± 2.0		0.059	
Plasma ascorbate (µmol/L)		57.8 ± 11.0		90.5 ± 55.5		0.017	
Plasma MDA levels (µmol/mL)		17.0 ± 8.6		10.6 ± 3.8		0.031	
Plasma F2IsoPs (pg/mL)		16.9 ± 4.8		12.0 ± 4.3		0.012	
Phagocytosis (%)		20.4 ± 8.6		30.7 ± 11.4		0.012	
Oxidative burst (%)		5.9 ± 3.5		11.1 ± 4.5		0.006	
Cholesterol (mg/dL)		239.3 ± 60.1		187.9 ± 45.9		0.050	
Triglyceride (mg/dL)		168.8 ± 63.8		195.9 ± 100.1		0.217	
HDL (mg/dL)		44.9 ± 10.6		45.9 ± 15.0		0.735	
LDL (mg/dL)		164.9 ± 53.7		126.5 ± 32.3		0.154	
Insulin (IU/mL)		14.1 ± 4.6		11.5 ± 3.6		0.057	
Notes.

Data represented as mean ± SD. P-value was calculated using paired t-test.

Abbreviations BMI body mass index

F2IsoPs F2-Isoprostanes

FBS fasting blood sugar

HbA1c glycated haemoglobin

HDL high-density lipoproteins

LDL low- density lipoproteins

MDA malondialdehyde

MiRNA screen revealed differences in the expression level of a subset of miRNAs at pre- and post-vitamin C supplementation

A screen of circulating miRNAs in the plasma of five subjects with T2DM at pre- and post-supplementation was performed using the NanoString platform. Of the 800 target miRNA in the screen, 26 miRNAs were expressed in all samples (Fig. S1 and Table S1). Compared to pre-supplementation levels, four miRNAs showed significantly different expression at post-supplementation. Of these, miR-451a was down-regulated post-supplementation (−1.72 fold change (FC), p = 0.036), while miR-1253 (0.62 FC, p = 0.027), miR-1290 (0.53 FC, p = 0.036) and miR-644a (0.5 FC, p = 0.042) were up-regulated post-supplementation (Fig. 1A). MiR-451a showed the highest reduced fold change of any of the other miRNAs tested. To validate these differences, samples from three additional subjects with T2DM were included in the qPCR analysis. Only miR-451a expression was found to be significantly different post-supplementation compared to the pre-supplementation timepoint (p = 0.030) (Fig. 1B). Using the same approach as above, we have recently demonstrated that there is no difference in the level of miR-451a in the plasma of placebo controls pre- and post-vitamin C supplementation (Chuangchot et al., 2020) (Fig. S2).

Figure 1 Patterns of miRNA expression.

(A) Analysis of circulating miRNA expression combined with clinical measures and qRT-PCR data in response to vitamin C supplementation in T2DM subjects with poor glycemic control (n = 5). For the miRNA, black and red bubbles represent pre-and post-supplementation, respectively. The size of the bubbles reflects the count of the specific miRNA relative to other miRNAs. The laboratory measures and miR-451a qRT-PCR data are indicated as a heat map from minimum value to maximum value of each parameter independently of other parameters; yellow to blue represents the low to high range. Pre; pre-vitamin C, Post; post-vitamin C supplementation. Numbers represent subjects 1–5. (B) Plot represents relative expression of miR-451a as validated by qRT-PCR (n = 8).

Correlation between expression of circulating miRNA and clinical laboratory measures of T2DM

The expression of miR-451a was then correlated with clinical laboratory measures for the subjects with T2DM and included all data from both the pre-and post-vitamin C supplementation timepoints. Expression of miR-451a was negatively correlated with plasma vitamin C levels (r =  − 0.497, p = 0.049, Fig. 2A) but positively correlated with MDA (r = 0.584, p = 0.017, Fig. 2B), cholesterol (r = 0.564, p = 0.022, Fig. 2C) and LDL (r = 0.522, p = 0.037, Fig. 2D) levels, suggesting that this miRNA might play a role in oxidative status and lipid metabolism.

Figure 2 Correlation between miR-451a expression and laboratory variables.

(A) Vitamin C levels, (B) MDA, (C) Cholesterol and (D) LDL. Black and red represent pre-and post-supplementation samples, respectively.

Target gene prediction and pathway enrichment analysis

Bioinformatics analysis tools were used to identify putative target genes and pathways likely to be affected by miR-451a. A total of 88 target genes of miR-451a were retrieved from the analysis and used in the program DAVID to identify relevant KEGG pathways. The top enriched KEGG pathways are presented in Table 2. The target genes are involved in several signaling pathways including mTOR signaling pathway, PI3K-Akt signaling pathway, estrogen signaling pathway, hepatitis B, pathway in cancer and colorectal cancer, AMPK signaling pathway, FoxO signaling pathway as well as other signaling pathways that share target genes such as IKBKB, PIK3CA and AKT1 as shown in Table 2 (description of genes in Table S2).

Table 2 KEGG signaling pathways of miR-451a.

Term	Gene targets	P-value	FDR	
mTOR signaling pathway	IKBKB, STK11, PRKAA1, CAB39, PIK3CA, AKT1, BRAF, PIK3R1	1.35E−08	1.13E−06	
PI3K-Akt signaling pathway	IKBKB, ATF2, STK11, PRKAA1, PIK3CA, MYC, BCL2, LPAR1, AKT1, PIK3R1, IL6R	5.77E−08	1.62E−04	
Estrogen signaling pathway	ATF2, PIK3CA, GNAQ, MMP2, AKT1, PIK3R1, MMP9	7.81E−07	1.62E−04	
Hepatitis B	IKBKB, ATF2, PIK3CA, MYC, BCL2, AKT1, PIK3R1, MMP9	9.40E−07	1.62E−04	
Pathways in cancer	IKBKB, PIK3CA, GNAQ, MMP2, MYC, BCL2, LPAR1, AKT1, BRAF, PIK3R1, MMP9	1.98E−05	2.39E−04	
Colorectal cancer	PIK3CA, MYC, BCL2, AKT1, BRAF, PIK3R1	1.99E−05	2.39E−04	
AMPK signaling pathway	STK11, PRKAA1, CAB39, RAB14, PIK3CA, AKT1, PIK3R1	2.99E−05	3.14E−04	
FoxO signaling pathway	IKBKB, STK11, PRKAA1, PIK3CA, AKT1, BRAF, PIK3R1	4.70E−05	3.95E−04	
Signaling pathways regulating pluripotency of stem cells	PIK3CA, PCGF5, MYC, AKT1, PIK3R1, BMI1	6.58E−05	5.02E−04	
Non-alcoholic fatty liver disease (NAFLD)	IKBKB, PRKAA1, PIK3CA, UQCRQ, AKT1, PIK3R1, IL6R	1.39E−04	7.81E−04	
TNF signaling pathway	IKBKB, ATF2, PIK3CA, AKT1, PIK3R1, MMP9	2.04E−04	1.07E−03	
Insulin resistance	IKBKB, PRKAA1, PIK3CA, GFPT1, AKT1, PIK3R1	2.32E−04	1.15E−03	
Notes.

FDR, false discovery rate.

Discussion

Type 2 diabetes mellitus (T2DM) is a worldwide health problem and continues to increase in prevalence (Cho et al., 2018). Overproduction of ROS caused by high glucose levels and dyslipidemia in long-standing T2DM leads to increased oxidative stress and chronic inflammation. These features can result in diabetic complications and increased susceptibility to infections resulting in increased morbidity and mortality, especially in uncontrolled diabetes (Critchley et al., 2018). These pathophysiological processes could impact miRNA expression and as suggested by numerous studies, which have investigated several dysregulated miRNAs in T2DM, these molecules can be used as potential biomarkers for predictive, diagnostic and therapeutic procedures (He et al., 2017; Jimenez-Lucena et al., 2018). Furthermore, there is evidence that upregulation of miR-21 is associated with elevated ROS levels and reduced antioxidant responses in prediabetics and subjects with T2DM (La Sala et al., 2019), indicating miRNA may be linked with pathological processes of the disease. Interestingly, vitamin C supplementation in subjects with T2DM has shown positive effects on blood glucose and oxidative stress (Afkhami-Ardekani & Shojaoddiny-Ardekani, 2007; Mazloom et al., 2011).

We hypothesized that vitamin C supplementation could modulate miRNA expression and correlate with the blood parameters of subjects for which we observed improved plasma vitamin C levels, reduced oxidative stress and increased PMN function after receiving 1,000 mg of vitamin C daily for six weeks. To our knowledge, the present study is the first report to investigate the effect of vitamin C supplementation on miRNA expression in subjects with T2DM and with poor glycemic control. The results from the initial miRNA screen revealed that miR-451a was downregulated, with the highest negative fold change, after supplementation and this was validated using qRT-PCR. However, the three significantly upregulated miRNAs from the microarray screen did not show differences in expression pre- and post-supplementation in the qRT-PCR validation step that also utilized additional samples. The discrepancy between the two platforms likely reflects the low template of these miRNAs and as such we focused only on miR-451a for further analysis.

MiR-451a (or miR-451, miRBase 22) is dicer independent and can be transcribed into a hairpin structure and then processed with the Argonaute 2 (Ago 2) protein (Wang, Wu & Yu, 2019). This miRNA is stable in peripheral blood and can be found in erythrocytes, PMN, mononuclear cells and platelets (Ghai et al., 2019; Masaki et al., 2007). MiR-451a has been reported in the blood circulation of elderly subjects with T2DM and is up-regulated in plasma samples of subjects with diabetic nephropathy as well as in the serum of subjects with acute diabetic Charcot foot, indicating the expressed miRNA may play a role in pathological processes of the disease (Catanzaro et al., 2018; Pasquier et al., 2018; Sayilar et al., 2016). In this study, the expression of miR-451a was negatively correlated with vitamin C levels but positively correlated with MDA levels, a marker of oxidative stress, suggesting a possible role in the oxidative status of the subjects.

MiR-451a is also important in the erythroid lineage and plays a role in ROS production in erythrocytes by targeting 14-3-3 zeta via the inhibition of FoxO3 (Yu et al., 2010). High oxidative stress may affect miR-451a production as a study from Ranjan et al. found that deficient miR-451a expression was associated with defective ROS generation due to reduced Ago2 protein levels in macrophages (Ranjan et al., 2015). Inhibition of miR-451a also reduced ROS, lipid peroxidation and DNA damage (Zhu et al., 2018). These data suggest that miR-451a expression may positively correlate with ROS production. In this study, miR-451a expression showed a positive correlation with cholesterol and LDL. These factors may exacerbate the progression of the disease accompanied with miR-451a expression. However, further work is needed to confirm the correlation of miR-451a with oxidative stress as found in the current study.

We also investigated the target genes of miR-451a and their functional enrichment using experimentally validated datasets. The target genes were significantly involved in signaling pathways that play an important role in cellular functions of biological processes as shown in Fig. 3 and Table 2. To support our results, studies in T2DM cardiomyopathy-induced mice with high ROS production displayed up-regulation of miR-451a that directly targeted calcium-binding protein 39 (CAB39) and resulted in down-regulation of the LKB1/AMPK signaling pathway. These target genes are listed in Table 2. Furthermore, knockout of this miRNA showed the opposite effect (Kuwabara et al., 2015). Thus, the data suggests that miR-451a expression was induced by high oxidative stress leading to suppression of the AMPK pathway, a central energy-sensing of metabolic regulation, which was found to have reduced activity in insulin-resistant individuals (Li et al., 2019; Xu et al., 2012). The reduced ROS production found in the subjects with T2DM in this study after vitamin C supplementation and with down-regulation of miR-451a may be through CAB39/LKB1/AMPK signaling, as PI3K/AKT/mTOR signaling is the central pathway of glucose and lipid metabolism (Huang et al., 2018; Tuo & Xiang, 2018). Hyperglycemia and excess FFAs lead to insulin resistance and high levels of ROS resulting in impaired PI3K/AKT/mTOR signaling found in subjects with T2DM (Huang et al., 2018). Moreover, reduction of oxidative stress has been reported in subjects with up-regulation of AKT and mTOR after pretreatment with vitamin C (Lin et al., 2016). MiR-451a could target multiple genes involved in the PI3K/AKT/mTOR signaling pathway including AKT1, PIK3CA, PIK3R1, and PRKAA1 (as indicated in Table 2 and Table S2). In addition, IKBKB is one of the target genes of miR-451a (Li et al., 2013) and is a known inhibitor of the NF-kB complex that can be activated by multiple harmful cellular stimuli such as stress or free radicals. There is evidence that vitamin C can also diminish NF-kB activation by directly inhibiting IKK β, aside from quenching ROS (Carcamo et al., 2004). The miRNA may be involved in this synergistic effect of vitamin C in response to ROS. Supplementation with vitamin C in the subjects might help to promote these pathways through the regulation of miR-451a. Further studies are required to explore the mechanisms behind the effects of vitamin C supplementation on this circulating miRNA associated with systemic oxidative stress in the subjects.

Figure 3 Proposed mechanism of the effect of miR-451a in response to vitamin C supplementation.

Red lines represent the results of this study and blue lines indicate potential mode of interaction for miR-451a. Poor glycemic controlled T2DM subjects receiving vitamin C 1,000 mg daily for six weeks showed significant down-regulation of circulating miR-451a accompanied with increased levels of vitamin C, reduced oxidative stress (MDA and F2IsoPs) and increased PMN function (phagocytosis and oxidative burst). Changes in oxidative status may modulate miR-451a expression by altering Ago2 protein and/or increased involvement of signaling pathways such as AMPK signaling by repressing its target genes. MiR-451a may play a role in neutrophil chemotaxis by targeting RAB5A and 14-3-3zeta resulting in the activation of p38 MAPK signaling. However, the role of this miRNA in phagocytosis and oxidative burst of neutrophils is still to be explored (dotted line) and may be reflected in intracellular miRNA changes rather than circulating changes as identified in this study.

The participants in this study showed enhanced phagocytosis and oxidative burst after supplementation of vitamin C, which is consistent with previous studies that have shown the effect of vitamin C in enhancing PMN function such as motility, chemotaxis, phagocytosis and microbial killing (Bozonet et al., 2015; Carr & Maggini, 2017). However, knowledge of the molecular mechanisms of the effect of vitamin C on these processes, especially phagocytosis and oxidative burst, are limited. MiR-451a has been reported to impact neutrophil chemotaxis by suppressing p38 MAPK through targeting Rab5a and 14-3-3 zeta (Murata et al., 2014). Rab5a, a small GTPase, regulates intracellular membrane trafficking involved in phagolysosome fusion during bacteria phagocytosis of neutrophils (Perskvist et al., 2002). Moreover, treatment of human bladder cancer cells by vitamin C could suppress p38 MAPK activity and ROS production (Kim et al., 2008). As suggested from these data, vitamin C may affect miR-451a expression and associate with neutrophil function through targeting Rab5a and the p38 MAPK pathway. However, investigation of intracellular miRNAs in response to vitamin C supplementation may help us to better understand the molecular mechanism underlying PMN function that we clearly observed in these subjects. A proposed mechanism of action for miR-451a in response to vitamin C supplementation is shown in Fig. 3.

Limitations of this pilot study include a small sample size. Increasing the number of the participants may better clarify the association of the candidate miRNA and the laboratory measures including improved oxidative status and PMN functions as well as reduced cholesterol after vitamin C supplementation. Nevertheless, the power of the test based on the difference between the two dependent groups using the mean and SD. of relative miR-451a data revealed that the effect size was 1.15 and the power of the test was approximated at 80%. Furthermore, the recent study compared only the miRNA expression between the pre-and post-vitamin C supplementation. However, our previous study of subjects receiving placebo showed no difference in miR-451a expression at pre-and post-supplementation as represented in Fig. S2. More research is needed to confirm and validate the finding that vitamin C intake modulates circulating miR-451a expression and to determine the functional role of this miRNA in subjects with uncontrolled T2DM.

Conclusions

In conclusion, this study has identified a novel association between vitamin C supplementation for six weeks and reduced circulating miR-451a expression in subjects with poorly controlled T2DM. This candidate miRNA may be used as a biomarker to identify subjects that respond to vitamin C treatment or oxidative status in plasma.

Supplemental Information

Supplemental Information 1 miRNA analysis with p-value

miRNA analysis showing 38 first rank with p-value.

Click here for additional data file.

Supplemental Information 2 GEO submission data

Data of GEO submission (GSE154647).

Click here for additional data file.

Supplemental Information 3 Raw data of miRNA count

Number of count of 813 miRNA found in samples.

Click here for additional data file.

Supplemental Information 4 Log2 normalized count

Data of log2 normalized count of all miRNA found in samples

Click here for additional data file.

Supplemental Information 5 qPCR data of mi-1253 and miR-1290

The data showed no significant difference of both miRNAs.

Click here for additional data file.

Supplemental Information 6 Supplemental Data

Figure S1: Number of baseline miRNAs from plasma of T2DM subjects. The numbers of baseline miRNAs are depicted on the y-axis and number of counts on the X-axis. The most miRNA counts are around 10–50.

Figure S2: miR-451 expression in subjects receiving placebo control (n = 8). The plot represents relative expression of miR-451a as validated by qRT-PCR using the cel-miR-39 spike-in control as a reference gene. Data is from eight subjects from a randomized, placebo controlled, crossover design study as has been reported recently (Chuangchot et al., 2020), who have received placebo daily for six weeks and were also investigated for miR-451a expression at the pre-and post-placebo supplementation. There was no significant difference after placebo supplementation (p = 0.308).

Table S1 : List of abundant plasma microRNAs at baseline of T2DM subjects

Table S2 : Common validated target genes of miR-451a involved in biological processes

Table S3 : Characteristics of the participants at the pre-and post-vitamin C and placebo supplementation timepoints.

Click here for additional data file.

Supplemental Information 7 Miame checklist

Details of the project followed the Miame checklist.

Click here for additional data file.

Supplemental Information 8 Raw data of qRT-PCR

Raw data of the validated qRT-PCR of miRNAs in 8 participants.

Click here for additional data file.

We would like to thank Exercise and Sport Sciences Development and Research Group (ESRG), Khon Kaen University for laboratory data collection used in this work.

Additional Information and Declarations

Competing Interests

Author Contributions

Human Ethics

Data Availability

The authors declare there are no competing interests.

Laongthip Ruknarong conceived and designed the experiments, performed the experiments, analyzed the data, prepared figures and/or tables, authored or reviewed drafts of the paper, and approved the final draft.

Chongchira Boonthongkaew and Nisa Chuangchot performed the experiments, analyzed the data, authored or reviewed drafts of the paper, and approved the final draft.

Amonrat Jumnainsong, Naruemon Leelayuwat, Apinya Jusakul and Chanvit Leelayuwat conceived and designed the experiments, analyzed the data, authored or reviewed drafts of the paper, and approved the final draft.

Silvana Gaudieri conceived and designed the experiments, analyzed the data, prepared figures and/or tables, authored or reviewed drafts of the paper, and approved the final draft.

The following information was supplied relating to ethical approvals (i.e., approving body and any reference numbers):

The Human Ethical Committee of Khon Kaen University approved this study (HE571264).

The following information was supplied regarding data availability:

Data is available at GEO: GSE154647.

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
