# Peer review of "Vitamin C supplementation reduces expression of circulating miR-451a in subjects with poorly controlled type 2 diabetes mellitus and high oxidative stress"

_PeerJ, doi:10.7717/peerj.10776_

## Round 0.1 · original submission · Minor Revisions

This study addresses a relevant and interesting topic. However, there are important gaps in the study that need to be revised before sending out for peer review. For example, miR-451a is modified in poorly controlled T2D patients, what happened with the controls? Is this a specific pattern in T2D patients or a general response to vitamin c consumption? If no controls are used, then the title should be modified accordingly. The material and methods section does not indicate clearly the number of participants. It is not clear for this editor if the n=8 subjects in the validation phase are the same from the screening. Please, better describe this section and include all necessary information in the manuscript before sending out for peer review.

---

## Round 0.2 · Major Revisions

The miR-451a bioinformatic analysis should be performed using only well validate targets from the literature, otherwise, the authors should perform a luciferase assay in order to demonstrate that a specific 3'UTR of mRNA target gene binds with miR.

·

Basic reporting

No comment

Experimental design

No comment

Validity of the findings

No comment

Additional comments

- Your introduction needs a reference to the end of the sentence on line 54/55;
- In line 88 of the introduction write what is the acronym PNM;
- Put at the bottom of table 1 what are the acronyms mentioned in the table.

·

Basic reporting

No comment

Experimental design

SAMPLE SIZE: The research has a very small sample size. What was the reason why the proponents didn’t consider a larger sample size to make generalizations more reliable? The sample size was based on what computation?
SAMPLE SELECTION: The inclusion criteria were mentioned, but what about any exclusion criteria that were considered? What possible biases were committed in the selection of samples? How did the authors address such possible biases? Do the authors see the small sample size as a factor that may have prevented the determination of inter- and intra-individual variability.
LACK OF CONTROL: What research design was employed in the study? Observational, cohort study design, or experimental study design? If the former was used, there shouldn’t be an intervention, while if it was latter, a control would have been necessary.It will be interesting to note how the supplementation would have influenced the miRNAs in a non-diabetic.
METHODOLOGY: There was no discussion about how the following parameters were determined: phagocytosis/ PMN function and oxidative burst, and reduced oxidative stress. Were the parameters measured only during pre- and post- intervention (supplementation with Vitamin C)?

Validity of the findings

The authors were able to establish the novelty of this paper in terms of investigating the influence of Vitamin C supplementation to the expression of miRNAs in T2DM.
However, the authors have conflicting statements in the following lines.
In the lines 291-293, it was mentioned that there's a need to confirm and validate using a control group, however, this is in contrast to the conclusion which said that miR-451a might be used as a biomarker. With a small sample size and lack of control, such conclusion may be inconclusive.

Reviewer 3 ·

Basic reporting

Dear authors,
the manuscript is unintelligible in some parts of the text, in particular when a tentative explanation of the experimental design is presented. Besides these aspects, the general organization and structure appear well done.
The background is poor in the context of diabetes and their pathogenesis and complications, especially in the explanation of the relationship between hyperglycemia and oxidative stress/antioxidant response, given the Vit-C a powerful antioxidant. So, I suggested in the file uploaded to introduce some references peculiar for framing miRs and their roles in metabolic derangement.
Regarding table and figures, they are kind, but lack of a clear explanation in discussion section.
Another important aspect is the validation of the miR-target gene. The authors should perform a luciferase assay in order to demonstrate that a specific 3'UTR of mRNA target gene binds with miR.

Experimental design

The experimental design would be ameliorated in the form of writing.
The diagnosis of diabetes should be described in the text, as well as the drug administered and the duration of the disease.

Validity of the findings

This argument is a vest but with a note of freshly due to miRNA approach. The introduction in this work of a time-point is very interesting, and add much information about the controlling mechanism of Vit-C on mirna expression in diabetics.

Annotated reviews are not available for download in order to protect the identity of reviewers who chose to remain anonymous.

---

## Round 0.3 · Minor Revisions

Because of the low number of subjects analyzed, this study should be considered as a pilot study. Please include at least in the abstract and the text that this is a pilot study. This is mandatory for final acceptance.

Reviewer 3 ·

Basic reporting

Please, check English by native speaker

Experimental design

Experimental design should be improved

Validity of the findings

without a validation of a target for mir-451a the manuscript appears only descriptive

Additional comments

Apart from the consideration that the authors did not consider my comment about the need to perform Luciferase assay, some parts of the experimental design are incomplete. For example, statistics to assess the effective correlation between glycemic parameters and mir-451a did not appear in the manuscript. So that, this miRNA not influence glycemic status, therefore diabetes status. However, also a statistical model more structured should ameliorate the analysis to find more robust data to present.
The levels of miR-451a were reduced in Vit-c supplementation, but the correlation shown in table 1 appeared borderline.
I recommend to seriously consider to identify a target to focalize the entire paper.

---

## Round 0.4 · accepted · Accept

The authors have revised the manuscript according to the reviewer suggestions.